# The Effectiveness of Interventions for the Prevention or Treatment of Paternal Perinatal Anxiety: A Systematic Review

**DOI:** 10.3390/jcm11226617

**Published:** 2022-11-08

**Authors:** Michelle L. Fisher, Paul Sutcliffe, Charlotte Southern, Amy L. Grove, Bee K. Tan

**Affiliations:** 1Division of Health Sciences, Warwick Medical School, Gibbet Hill Campus, University of Warwick, Coventry CV4 7AL, UK; 2Department of Cardiovascular Sciences, University of Leicester, Leicester LE1 7RH, UK

**Keywords:** father, paternal, anxiety, perinatal, intervention, systematic review

## Abstract

**Background:** The worldwide prevalence of paternal perinatal anxiety (PPA) ranges between 3.4% and 25.0% antenatally, and 2.4% and 51.0% postnatally. Experiencing PPA can adversely impact the individual, partners, and infants. Research concerning PPA is lagging and fragmented compared to research for new mothers. **Objectives:** To establish the effectiveness of prevention or treatment interventions for PPA in adults identifying as male. **Data sources:** We completed searches of Medline, EMBASE, PsycINFO and Web of Science from inception to 2 December 2021, as well as hand searches of references from relevant papers. **Search selection and data extraction:** Randomised controlled trials delivering prevention or treatment interventions and reporting anxiety outcomes for new/expectant fathers in the perinatal mental health period were included. Our review follows the PRISMA reporting guidelines. One reviewer independently screened 5170 titles/abstracts; second reviewers screened 50%. Two reviewers independently screened full text, extracted data, and conducted risk of bias assessments. **Synthesis:** Cochrane’s collaboration tool 2 was used to assess quality. Primarily results are synthesised narratively, a post-hoc sub-group analysis was completed on four studies using the same outcome measure. **Main results:** Twelve of the 5170 studies fulfilled the inclusion criteria. Studies used psychoeducational or practical skills interventions. Interventions mostly involved couple-dyads and three studies assessed PPA as a primary outcome. Included interventions were prevention-based; no treatment interventions were found. Father-only interventions consistently reported a significant reduction of PPA. **Conclusions:** Systematic searching yielded no treatment interventions, highlighting a substantial gap in the evidence base. Within a limited and heterogenous sample, no studies targeted diagnosed PPA. Evidence suggested father-focused interventions may be effective in preventing PPA, regardless of the intervention delivery mode or intervention content. However, consistency between study design and options within the field are lacking compared to interventions available for mothers.

## 1. Background

The perinatal mental health period (PMHP), from early pregnancy up to 12 months after birth [1], is a time of increased risk for developing mental health illness, commonly anxiety and depression. Compared to perinatal mental health research for mothers, paternal mental health is an understudied area; five Cochrane reviews have appraised the effectiveness of perinatal interventions for mothers [2,3,4,5,6]. This may be due to recruitment logistics, whereby fathers are recruited to studies developed for mothers [7], as opposed to studies targeted at fathers, for which a clinical population may be harder to distinguish.

The transition to fatherhood is recognised as a ‘developmental milestone’ [8], and emerging PMHP literature recognises fatherhood as a time of significant change encompassing a number of major stressors [8], and increased risk of mental health issues [7] that can be unrelated to maternal wellbeing [9,10]. Office of National Statistics (ONS) data found men under 50 account for 75% of deaths by suicide in the UK [11], a demographic which includes a proportion of new/expectant fathers.

From a basic science perspective, male animal models show distinct neurohormonal changes during the perinatal period due to hormonal fluctuations and structural changes triggered by parenthood [12]. Animal models of anxiety show a distinct sex-related dimorphism in neurophysiological functioning with the male sex more susceptible to changes in the hypothalamus–pituitary–adrenals (HPA) axis [13].

Worldwide prevalence of paternal perinatal anxiety (PPA) broadly ranges from 3.4% to 25% antenatally and 2.4% to 51.0% postnatally [14], with discrepancies between prevalence rates due to underreporting and variance between outcome measures. Perinatal anxiety is not defined in The Diagnostic and Statistical Manual of Mental Disorders, Fifth Edition (DSM-V) [15], however PPA is linked to adverse outcomes for the entire family including poor mental health for both parents, relationship dysfunction, disordered attachments and, short and long-term child development issues [16,17,18,19,20,21,22]. For this review PPA is classified as ‘generic anxiety symptomatology’ using Philpott and colleagues’ criteria [14]; anxiety is situational and does facilitate safety seeking behaviours, however transient anxiety experienced through fatherhood can disrupt quality of life [23]. It is common for new fathers to experience sub-clinical levels of anxiety, and therefore self-report measures are common assessment tools.

Given the challenges faced by new fathers, the likelihood of developing PPA, and the potential perinatal neurological differences in males as a sex, there is a requirement for further interrogation of the evidence base. Previous systematic reviews of paternal mental health interventions have focussed on couple-dyads [24], the antenatal period [25], paternal perinatal depression [26], and a mini-review has considered PPA and depression [27], as there often exists a comorbidity [28]. Most recently, Goldstein observed heterogeneity between research designs and a lack of research regarding diagnosed symptoms of depression [26]. Similarly, Rodrigues found no treatment of clinically diagnosed depression or anxiety [27], however no formal quality assessment was completed.

Elaborating upon previous reviews, with particular emphasis on Goldstein’s systematic review of interventions for paternal perinatal depression [26] and Rodrigues’ mini-review involving PPA [27], this review aims to systematically appraise the efficacy of prevention or treatment interventions on PPA outcomes in adults identifying as male over the age of 18, compared to standard care. This review will identify, and quality assess randomised controlled trials (RCTs), the gold-standard of primary research designs, reporting PPA symptomatology as primary or secondary outcomes to maximise a yield of scientifically rigorous results.

## 2. Methods

### 2.1. Aims

Our primary aim is to review the published evidence to assess the effectiveness of prevention or treatment interventions on PPA outcomes in adults identifying as male over the age of 18. Patients were not involved in developing this review.

The population, intervention, comparators, outcomes, study design (PICOS) has been used to explicate the research question into concepts and search terms, outlined in Table 1. In the context of this review PPA considers any anxiety disorder or dysfunction—classified as prolonged feelings of worry, nervousness and/or panic [29]. A core outcome set (COS) developed for maternal perinatal depression treatment is considered as a secondary outcome [30].

### 2.2. Registration and Reporting

The study protocol was registered in the international prospective register of systematic reviews (PROSPERO; registration number: CRD42020165609) and is reported using the Preferred Reporting Items for Systematic Review and Meta-Analysis (PRISMA) statement (see Figure 1) [31].

### 2.3. Search Strategy

A MEDLINE search strategy (see Figure 2) was used and adapted with the support of an information scientist for the following three databases; EMBASE, PsycINFO and Web of Science. Databases were searched from the first available date to 2 December 2021. Deliberately inclusive search terms were used to capture all interventions, regardless of the country of origin. Searches were not restricted by date or language; one paper was produced in German [32] and screened by a colleague who is fluent. We also completed citation searches of included papers and key systematic reviews, hand searches of grey literature and the reference lists of relevant review studies.

Title and abstracts were screened against the PICOS inclusion criteria (see Table 1) by the first reviewer (MLF) on EndNote (Clarivate Analytics, EndNote X9.3.3, 2020). 20% of the original search results (3 February 2020) were subsequently screened by one reviewer (PS) and 30% screened by a visiting researcher. The first reviewer (MLF) and a second reviewer (CS) independently screened all title and abstracts of updated searches (6 January 2021, 2 December 2021). Included papers were screened by their full text independently by two reviewers (MLF & CS). Disagreements were adjudicated by a third reviewer (PS).

### 2.4. Risk of Bias

Two reviewers (MLF & CS) carried out risk of bias assessments using Cochrane’s Collaboration Tool 2 for assessing risk of bias [33]. The risk of bias is scored as high, medium, or low, and includes: selection bias, performance bias, detection bias, attrition bias, and reporting bias [33]. Discrepancies were resolved through adjudication (PS). Studies were not excluded on risk of bias assessments.

### 2.5. Data Synthesis

Meta-analysis of the primary outcome paternal perinatal anxiety was planned. We anticipated a high level of heterogeneity in the included studies; studies were post hoc sub-grouped by outcome measure where possible. All studies were analysed narratively by the first reviewer (MLF) and checked by two second reviewers (ALG & BKT). Narrative synthesis follows the PICOS format.

We performed random effects meta-analysis (performed by MLF and checked by a statistician) for psychoeducational interventions using State-Trait anxiety inventory (STAI) [34], a validated outcome measure assessing state and trait anxiety;

State anxiety: acute anxious feelings resulting from action, thought, or environment; more likely to be experienced by new parents [35].

Trait anxiety: personality related, experienced regularly rather than situationally [34].

## 3. Results

### 3.1. Study Characteristics

Twelve studies are included in this review, six of which were also identified within Goldstein’s [26] systematic review of paternal perinatal depression [28,36,37,38,39,40]. Study characteristics are detailed in Table 2. Up to 2 December 2021, 5170 studies were identified. Nine studies were delivered in high-income countries: USA [36,37,41,42], Australia [28,39], Taiwan [43], Singapore [44], Netherlands [45]. Three studies were delivered in upper-middle-income countries: China [40] and Iran [38,46].

#### Patient Demographics

Across twelve studies, 1921 male participants took part. The average age of participants ranged from 27.9–35.3 years in intervention groups, and 27.9–34.73 in control groups. Further, 85–98% of participants were full-time employed. In four studies, all participants were married [38,42,44,46], and in eight studies 87–100% of participants were married or cohabitating [28,36,37,39,40,41,43,45].

### 3.2. Interventions

All interventions are prevention-based, no treatment interventions were found. Intervention models follow two categories:psychoeducational interventions *n* = 9 [28,36,38,39,41,43,44,45,46].practical skills interventions *n* = 3 [37,40,42].

#### 3.2.1. Psychoeducational

Nine studies adopted a psychoeducational model [28,36,38,39,41,43,44,45,46]; interventions combined educational and psychological approaches; consisting of either child-birthing [28,43,46] or child-rearing education [36,38,39,41,44,45], delivered alongside psychological support regarding; new parent lifestyle [38,44], co-parenting/conflict resolution [36,41], or strategies for managing emotional wellbeing [28,38,39,43,45,46]. Four interventions were delivered through tele- [38,39,44,45] or mobile App [44].

#### 3.2.2. Practical Skills

Three studies utilised interventions teaching fathers new parenting or pregnancy-related skills to practice [37,40,42]. All interventions were delivered face-to-face, and one had an additional technological element, a training DVD [37]. These interventions can be subdivided into two categories:infant-focused: new fathers practicing skin to skin contact (SSC) [40] with their baby.partner-focused: expectant fathers practicing pregnancy massage [37,42] or relaxation techniques [42].

### 3.3. Comparators

Nine studies reported standard care provision or usual care as the comparator [37,38,39,40,41,42,43,44,46]. Three studies reported the comparator as: pre-existing antenatal education [28]; childcare provision leaflet [36]; waitlist control group [45].

### 3.4. Outcomes

Outcomes are reported narratively (Table 3). PPA outcomes are documented as primary (*n* = 3) or secondary outcomes (*n* = 9). Seven studies reported ‘paternal health’ as the primary outcome [28,38,39,40,42,43,46]. Four studies reported ‘parental health’ as their primary outcomes [36,41,44,45]. One study reported ‘maternal health’ as the primary outcome [37]. A post-hoc sub-group meta-analysis was performed on psychoeducational interventions reporting outcomes using the STAI, with individual Trait and State I^2^ of 78% and 15%, respectively.

Across the twelve studies, five validated outcome measures were used to record PPA:State-trait anxiety inventory (STAI) [34].Zung self-rating anxiety scale (SAS) [47].Depression anxiety and stress scale (DASS) [48].Hospital anxiety and depression scale (HADS) [49].Taylor manifest anxiety scale [50].

We also extracted data relating to the treatment-focused core outcome set [30], however interventions identified are prevention-based.

#### 3.4.1. Psychoeducational Intervention Outcomes

All group-delivered couple-dyad interventions reported no significant intervention effect on PPA [28,36,39,41]; Baby Triple P parenting education program, delivered face-to-face with telephone-consultation, reported no significant effect on PPA post-intervention, or at six-months follow-up (time effect, B = −0.20, F (1, 85.25) = 2.59, *p* = 0.111; group effect, B = −0.16, F (1108.44) = 0.14, *p* = 0.709) [39]. Antenatal education sessions led by a male facilitator had no significant effect on PPA, although some difference favouring the experimental arm were reported (participant reported reduction N(%) 6-week postnatal; intervention = 36(12.4%), control = 28(11.4%)) [28]. Two studies were first [36] and second [41] iterations of ‘family foundations’. Both iterations reported no significant effect on PPA ten-months postnatal (intervention = M 16.83 SD (4.52), control = M 17.62 SD (5.4)) [41], or six-months postnatal (B = 0.816, SE = 0.51, *p* > 0.1) [36]. However, a significant self-reported effect on maternal depression (effect size 0.56 (b = −1.95 < 0.01)), anxiety (effect size 0.38 (b = −1.218 < 0.05)) [36], and infant soothability (b = 0.19; *p* < 0.05) [41] suggested family foundations was not effective for fathers specifically.

Two interventions were delivered to individual couple-dyads [44,45];

Educational child-rearing and lifestyle App reported significant effect on PPA compared to control, at one-month (OR = −3.40, 95% CI (3.93 to −2.86), *p* < 0.001), and three-months postnatal (OR = −1.09, 95% CI (−1.57 to −0.61), *p* < 0.001) [44].Face-to-face child-rearing education and telephone emotional wellbeing support reported no significant effect (ten-weeks postnatal = intervention *M* 5.64 *SD* (2.77); control *M* 5.50 *SD* (2.56)) [45].

Father-only psychoeducational interventions consistently reported significant intervention effects on PPA [38,43,46]; Perinatal counselling intervention significantly reduced state anxiety four-weeks post-intervention (MD: −2.4; 95% CI: −4.7 to −0.2; *p* = 0.03) [46]. Charandabi and colleagues’ antenatal lifestyle education combined with telephone-consultations reported significant intervention effect on PPA six-weeks postnatal (State Anxiety = OR −7.5 95% CI (−11.6 to −3.4), Trait Anxiety = OR −8.3 95% CI (−12.2 to −4.4) [38]. Li and colleagues’ childbirth education program [43] significantly reduced PPA (F = 3.38, *p* = 0.001) when implementing analysis of covariance.

#### 3.4.2. Practical Skills Intervention Outcomes

All practical skills interventions, delivered individually to couple-dyads [37,42] or fathers-only [40], reported a difference in self-reported PPA between groups, favouring the intervention arms. SSC between father and infant significantly reduced self-reported PPA (t = −1.321, *p* < 0.05) [40], compared to care as usual.

Two partner-focused interventions reported a difference between groups:Pregnancy massage therapy (t = 3.61, *p* < 0.01) reduced self-reported PPA compared to care as usual [37].Antenatal partner massage or relaxation interventions outcomes showed a negative relationship from baseline to five-weeks post-intervention, compared to care as usual (r = 0.31, *p* < 0.01) [42]. At five-weeks post-intervention, correcting for multiple testing revealed partner massage was significantly effective in reducing PPA compared to relaxation (4.03, *SE* = 0.78, *p* = 0.001) [42].

#### 3.4.3. Technology

Five interventions utilised technology for delivering interventions [37,38,39,44,45]. Two psychoeducational interventions using telephone-consultation in combination with face-to-face delivery reported no significant effect on PPA; (time effect, B = −0.20, F (1, 85.25) = 2.59, *p* = 0.111; group effect, B = −0.16, F (1108.44) = 0.14, *p* = 0.709) [39]; (ten-weeks postnatal = intervention *M* 5.64 *SD* (2.77); control *M* 5.50 *SD* (2.56)) [45]. Three interventions reported a significant effect on PPA:Psychoeducational intervention using telephone-consultation and App education (one-month postnatal (OR = −3.40, 95% CI (3.93 to −2.86), *p* < 0.001); three-months postnatal (OR = −1.09, 95% CI (−1.57 to −0.61), *p* < 0.001)) [44].Psychoeducational telephone-consultation (State Anxiety = OR −7.5 95% CI (−11.6 to −3.4), Trait Anxiety = OR −8.3 95% CI (−12.2 to −4.4)) [38].Pregnancy massage therapy: training delivered by DVD (t = 3.61, *p* < 0.01) [37].

**Table 3 jcm-11-06617-t003:** Outcomes.

First Author/Ref	Intervention Type	N Used at Final Analysis	Delivery Time	Timepoints	Outcome Measure	Intervention Outcomes
Psychoeducational						
*Fathers only*						
Charandabi (2017)/[38]	(prenatal lifestyle-based education)	N = 125 IG = 62 CG = 63	Antenatal to postnatal	T1—Baseline (24–28 weeks gestation) T2—8 weeks post training (32–36 weeks gestation) T3—6 weeks postnatal	STAI	**Treatment effect** Adjusted odds ratio, Baseline (24–28 weeks gestation)—State Anxiety = OR 1.1 95% CI (−2.1 to 4.5) Trait Anxiety = OR 2.7 95% CI (−1.6 to 6.5); T2 (8 weeks post training)—State Anxiety = OR −5.7 95% CI (−8.6 to −2.9) Trait Anxiety = OR −5.0 95% CI (−7.8 to −2.2); T3 (6 weeks postnatal)—State Anxiety = OR −7.5 95% CI (−11.6 to −3.4) Trait Anxiety = OR −8.3 95% CI (−12.2 to −4.4) ** **Narrative report** Compared with the control group, there was a significant reduction in self-reported state and trait anxiety scores at 8 weeks post intervention. Implementation of antenatal training interventions was reported as being easy.
Li (2009)/[43]	Birth Education Program for Expectant Fathers Who Plan to Accompany Their Partners Through Labour (childbirth classes)	N = 87 IG n = 45CG n = 42	Antenatal	T1—Baseline (34–36 weeks gestation) T2—1 day postnatal	STAI	**Treatment effect** Postnatal state anxiety analysis of covariance, F = 3.38, *p* = 0.001 ** **Narrative report** No statistical significance was self-reported between the intervention and control groups of fathers in trait anxiety. ** When analysis of covariance was implemented, correcting for education level, sources of childbirth information, attendance at Lamaze childbirth classes, and childbirth expectations at baseline, the effect of the intervention on postnatal state anxiety scores was significant.
Mohammadpour (2021)/[46]	Counselling sessions (4 weeks) to familiarize fathers with changes in pregnancy and their role in maternal and foetal health.	N = 102 IG n = 51 CG n = 51	Antenatal	T1—Baseline (20–24 weeks gestation) T2—4 weeks post-intervention	STAI	**Treatment effect** MD: −2.4; 95% CI: −4.7 to −0.2; *p* = 0.03 ** **Narrative report** State anxiety in the intervention group decreased significantly 4 weeks after the intervention compared to the control group. No significant difference was found between the two groups for trait anxiety.
*Group-based couple dyads*						
Feinberg (2016)/[41]	Family foundations (parenting education program)	N = 608 (women = 304, men = 304) IG = 152 CG = 152	Antenatal to postnatal	T1—Baseline T2—10 months postnatal	STAI	**Treatment effect** T2 (10 months postnatal) intervention, *M* 16.83 *SD* (4.52), control *M* 17.62 *SD* (5.4) **Narrative report** *Paternal* No significant intervention effects were found. *Infant *** Significant intervention effects were also reported for infant soothability; b = 0.19; *p* < 0.05, and reductions in sleep problems—difficulty falling back to sleep and number of wakings during the night—were reported.
Feinberg (2008)/[36]	Family foundations (parenting education program)	N = 304 (women = 152, men = 152) IG = 79 CG = 73	Antenatal to postnatal	T1—Baseline T2—6 months postnatal	Taylor Manifest Anxiety Scale	**Treatment effect** B = 0.816, *SE* = 0.51, *p* > 0.1 **Narrative report** *Paternal* No significant intervention effect on anxiety outcomes were self-reported *Maternal *** Significant intervention effects were described for maternal anxiety, and distress in the parent–infant relationship. *Infant ** Intervention group infants had a greater level of soothability by father report; B = 0.312, *SE* = 0.16, *p* < 0.1; effect size 0.35
Mihelic (2018)/[39]	Baby Triple *p* (parenting education program)	N = 224 (women = 112, men = 112), IG = 55 CG = 57	Antenatal	T1—Baseline T2—post-intervention T3—6 months postnatal	DASS-anxiety	**Treatment effect** Time effect, B = −0.20, F (1, 85.25) = 2.59, *p* = 0.111, Group effect, B = −0.16, F (1108.44) = 0.14, *p* = 0.709 **Narrative report** *Paternal* No significant intervention effects were self-reported at post-intervention or 6 months follow-up. Fathers in both intervention and control groups self-reported significant increases in their parenting confidence and self-efficacy. *Infant* No significant effect on father-infant bonding difficulties (*d* = 0.02, 95% CI (−0.35–0.39)), or parental responsiveness (*d* = −0.12, 95% CI (−0.49–0.24))
Tohotoa (2012)/[28]	(antenatal education)	N = 556 IG = 303 CG = 253	Antenatal to postnatal	T1—Baseline T2—6 weeks postnatal	HADS	**Treatment effect** Participants recording a reduction N(%), ~*p* is significant at 0.015 using McNemar-Bowker test, T2 (6 weeks postnatal) = intervention = 36(12.4%~), control = 28(11.4%) * **Narrative report** *Paternal* Fathers in the intervention group self-reported lower anxiety scores compared to fathers in the control group from baseline to post-intervention, however this was not a significant difference.
*Individual couple dyads*						
Missler (2020)/[45]	(psychoeducational parenting intervention)	N = 189 (women = 120, men = 69), IG = 31 CG = 38	Antenatal to postnatal	T1—Baseline (26–34 wks gestation) T2—34–36 weeks gestation T3—6 weeks postnatal T4—10 weeks postnatal	HADS	**Treatment effect** T4 (10 weeks postnatal) intervention *M* 5.64 *SD* (2.77), control *M* 5.50 *SD* (2.56) **Narrative report** *Paternal* No between group differences were observed on anxiety. The intervention was rated as useful by parents, and the information booklet was considered the most useful part of the intervention. Intervention and control groups showed an increase in self-reported distress after birth
Shorey (2019)/[44]	(technology-based supportive educational parenting program)	N = 236 (women = 118, men = 118) IG = 59 CG = 59	Antenatal to postnatal	T2—2 days postnatal T3—1 month postnatal T4—3 months postnatal	STAI	**Treatment effect** Adjusted odds ratio, T2 (2 days postnatal) OR = 0.13, 95% CI (−0.18 to 0.44), *p* = 0.4, T3 (1 month postnatal) OR = −3.40, 95% CI (3.93 to −2.86), *p* < 0.001, T4 (3 months postnatal) OR = −1.09, 95% CI (−1.57 to −0.61), *p* < 0.001 ** **Narrative report** *Paternal* The mean difference of self-reported anxiety scores were significantly lower at 1 and 3 months postnatal compared to the control group. *Maternal *** The mean difference of self-reported anxiety scores were significantly lower at 1 and 3 months postnatal compared to the control group.
**Practical skills**						
Field (2008)/[37]	(pregnancy massage therapy)	N = 114 (women = 57, men = 57) IG = 29 CG = 28	Antenatal	T1—Baseline T2—6 months postnatal	STAI	**Treatment effect** Independent samples *t* tests, t = 3.61, *p* < 0.01 ** **Narrative report** *Paternal* Fathers in the intervention group self-reported decreased anxiety, compared to the control group. *Maternal ** Mothers in the intervention group reported decreased depression, anxiety, and anger compared to the control group.
Huang (2019)/[40]	(paternal skin to skin contact)	N = 100 IG n = 50 CG n = 50	Postnatal	T1—Baseline T2—Directly after intervention	SAS	**Treatment effect** Independent samples *t* tests, t = −1.321, *p* < 0.05 ** **Narrative report** *Paternal* Fathers in the intervention group had significantly lower self-reported scores of anxiety and better role attainment than those in the control group. *Maternal *** Duration of breastfeeding after SSC in the intervention group was significantly longer than the control group. *Infant ** Infants in the intervention group had a more stable heart rate and forehead temperature, less duration of crying, and started feeding behaviour earlier.
Latifses (2005)/[42]	(1. relaxation) (2. partner massage)	N = 278 (women = 139, men = 139) IG1 = 46 IG2 = 47 CG n = 46	Antenatal	T1—Baseline T2—5 weeks follow-up	STAI	**Treatment effect** one way repeated measures analysis of variance T1-T2, Wilks’ = 0.77; F (2, 170) = 25.85, *p* = 0.001 * correlation coefficient, r = 0.31, *p* < 0.01 Bonferroni adjustment massage-relaxation, 4.03, *SE* = 0.78, *p* = 0.001 ** **Narrative report** *Paternal* Partner massage therapy lowered fathers’ self-reported anxiety levels. *Infant ** Fathers who reported low anxiety also reported high foetal attachment (this is a high predictor of infant attachment).

** significant treatment effect, * score difference favouring experimental, non-significant. IG = Intervention Group; CG = Comparator Group; OR = Odds Ratio.

#### 3.4.4. Core Outcome Set Findings

From the recent core outcome set developed for treatment interventions in perinatal depression [30], three core outcomes were reported by included studies:Self-assessed symptoms: All studies assessed outcomes through self-reporting, all outcome measures were validated for general population.Parent-infant bonding: Three interventions recorded outcomes for father-infant bonding.
I.Family foundations (second iteration) reported significant intervention effect on father-reported infant soothability (b = 0.19; *p* < 0.05) [41];II.Family foundations (first iteration) noted some intervention effect on father-reported infant soothability (B = 0.312, SE = 0.16, *p* < 0.1; effect size 0.35) [36];III.SSC intervention reported physical health differences in infants related to bonding and reduced stress—the intervention group recorded decreased infant heart rates (heartrate (bps) at 5 min/30 min (M(sd)): intervention = 145.61(3.21)/140.33(8.29); control = 146.07(4.83)/143.81(5.63)) [40].Adverse events and suicidal thoughts: Two studies considered adverse events [28,43].
I.Participants self-reporting anxiety scores at a cut-off suggesting severe anxiety were referred to a ‘clinical nurse specialist’ [28];II.Limiting participants to 3–4 couples to manage/deter adverse events in psychoeducational groupwork [43]. No studies considered the risk or management of suicidal thoughts.


### 3.5. Risk of Bias Assessment

The studies were low-medium risk of bias (see Figure 3). When separating studies by interventional approach, the risk of bias differed with psychoeducational interventions generally demonstrating lower risk of bias compared to practical skills interventions.

#### Post Hoc Subgroup Analysis

Four studies were included across two post hoc subgroup analysis [38,41,43,46]. All studies reported outcomes from both aspects of the STAI [38,43,46] or trait-only [41]. Random effect meta-analyses were performed on trait and state anxiety outcomes separately using RevMan software (Version 5.4.1. The Cochrane Collaboration, 2020) [51].


*Trait-anxiety*


We produced a random effects meta-analysis [52] to demonstrate the effectiveness of psychoeducational interventions in preventing trait-anxiety [34] (Figure 4). The effect of psychoeducational interventions averagely favours the experimental group over standard care provision. However, significant heterogeneity is present (four studies, 618 participants, Mean Difference = −2.34, 95% CI −5.40–0.72, *p* = 0.130; I^2^ = 78%). Findings were considered high certainty according to the Cochrane’s risk of bias assessment of the quality of evidence (Figure 5).


*State-anxiety*


Figure 6 reports the effect of father-focused interventions on state-anxiety. The effect of father-focused psychoeducational intervention was greater than standard care provision (three studies, 314 participants, Mean diff. −5.24, 95% CI −7.52–−2.96, *p* ≤ 0.00001; I^2^ = 15%). Findings were considered of high certainty according to the Cochrane’s risk of bias assessment of the quality of evidence (see Figure 7) and given the comparable methodology between studies (sample size, intervention models).

## 4. Discussion

Our aim was to systematically review relevant evidence to assess the effectiveness of prevention or treatment interventions on PPA outcomes in adults who identify as male over the age of 18, compared to standard care. All included studies were of prevention-based interventions; no treatment interventions were found. In addition, no studies targeted diagnosed PPA, and three interventions reported PPA as a primary outcome [38,43,46].

Our study addresses an important and prevalent issue; a large proportion of new/expectant fathers in the UK at risk of PPA (2.4–51%) [2]. Our findings have implications for clinical practice demonstrating a need for father-specific interventions. However, we are cautious in drawing firm conclusions given the paucity of studies. Other reviews of interventions have focussed on fathers as partners [24], antenatal educational interventions [25], or paternal perinatal depression [26]. Depression, a common perinatal mental health issue experienced by fathers, often co-exists with anxiety [53], yet it is only explored by one mini-review [27].

Within a small evidence base, we found that father-focused interventions (*n* = 4) i.e., delivered to fathers only, either as a group or individually, were effective in preventing PPA. Post-hoc subgroup analysis of the three psychoeducational interventions indicated the average effect [53] between father-focused psychoeducational interventions favoured the experimental arm (MD = −5.24, 95% CI −7.32–−3.17, *p* ≤ 0.00001; I^2^ = 15%). Father-focused practical skills (SSC between father and infant) also significantly reduced self-reported PPA (t = −1.321, *p* < 0.05) [40].

All three practical skills interventions reported a difference in self-reported PPA between groups and interventions delivered using technology were also proportionately effective in preventing PPA, three out of five reported a significant effect on PPA [37,38,44]. Group-delivered couple-dyad psychoeducational interventions were least effective in preventing PPA, consistently reporting no significant effects [28,36,39,41].

### 4.1. Interpretation

From a small evidence base, father-focused interventions were effective in preventing PPA; regardless of the intervention delivery mode; groupwork vs. individuals, or intervention content; psychoeducation vs. practical skills.

We postulate that the prioritisation of fathers, using father-focused methodologies, may be a reason for success. However, without including a sample of fathers experiencing PPA, it is difficult to explore this finding relative to the condition. The effectiveness of father-focused groupwork could stem from the strong body of evidence regarding the use of peer support to manage perinatal mental health issues [54]. This evidence is prominent for new or expectant mothers. However, the theoretical underpinnings of social inclusion transcend demographics. Integration within social networks where participants can find commonalities in experiences, particularly challenging ones such as parenthood, can create positive psychological impacts through shared understanding and a sense of belonging [55].

All three practical skills interventions reported a reduction in PPA. This is an important observation for a small evidence base. As highlighted through the Public Health Warwickshire ‘five ways to wellbeing’ initiative [56], it is widely known that learning a new skill contributes to improving wellbeing levels.

Digital technology also appears to play a key role in intervention success, with three interventions reporting an effect in preventing PPA. Fathers may respond well to technologically delivered interventions due to accessibility, considering typical fathering practices favouring work commitments over perinatal appointments. However, due to heterogeneity, these findings are interpreted cautiously.

Group-delivered psychoeducational interventions involving couple-dyads were least effective suggesting that groupwork with other couple-dyads are not conducive to an effective environment for mental health support for fathers. As suggested in earlier reviews [25,26], the reasons for this finding could stem from perceptions of the male role in the perinatal period as, traditionally, this has been female-focused. Perhaps the involvement of the father may not have been optimal.

### 4.2. Future Research

Our key finding was that father-focused interventions for PPA are most effective in reducing self-reported PPA. However, there was a small number (*n* = 12) of studies and variance between study methodologies. In addition, no participants were formally diagnosed with PPA or accessing interventions to treat PPA.

We have also highlighted a gap in the evidence base and a potential area to interrogate in future research. Treatment interventions could be informed by this review, other key review papers [24,25,26,27] and the core outcome set [30] for maternal mental health. This could improve consistency in recording, reporting and outcomes for mental health interventions.

In addition, we recommend authors consider patient safety more thoroughly in future research as this was not widely documented within our review. Patient safety is important to minimise unsafe practices which can be triggering and lead to adverse events such as suicide. The male adult population within the UK are most at risk of suicide.

#### Implications for Practice

We highlight the importance of paternal mental healthcare with emerging findings that a significant effect on reducing PPA was due to father-focused interventions and highlight the lack of focus on diagnosis and treatment. We suggest future perinatal care should have greater targeted involvement of the father to enable more positive perinatal mental health experiences.

### 4.3. Strengths and Limitations

To the best of our knowledge, this is the first systematic review on PPA specifically.

Our review can inform attempts to improve the effectiveness of interventions for a common perinatal mental health issue experienced by fathers. We have taken a systematic approach and found consistent differences in outcomes between different types of intervention, allowing us to draw distinct preliminary conclusions that can be expanded and tested.

Another strength in the findings of our review is recognising and advocating the significant role of fathers in pregnancy and highlighting the issue of perinatal mental health. Fathers are often a cornerstone in buffering maternal mental health, and we have accentuated the role that fathers could play in this respect. It is important to ensure that the population of fathers are recognised and supported to improve family mental health outcomes. Clinicians and healthcare providers cannot rely on new fathers as caregivers to their partners and babies without anticipating a need for support, like new mothers.

The main limitation is the small evidence base and study quality. Nine studies reported standard care provision or usual care as the comparator but did not specify what this level of care included [37,38,39,40,41,42,43,44,46]. PPA outcomes were self-reported by participants, no clinical interviews or prior diagnosis were required to participate, and there was no outcome to compare against. In addition, participants were not indicative of a general population with a third (*n* = 4) of studies solely including married participants.

## 5. Conclusions

Interventions for PPA are most effective in reducing self-reported PPA when they are father-focused. However, those targeted at group-based couple-dyads have the least impact on reducing PPA. Generally, interventions are well described, yet methodological heterogeneity exists within a small evidence base, making our findings preliminary conclusions that can be expanded and tested in future research.

The existing evidence base needs development to give fathers adequate recognition in perinatal healthcare, like mothers already have, and to reduce the likelihood of adverse events in early family life.

We recommend that strict methodological processes should be implemented in future research. Consulting the existing core outcome set would be most advantageous to produce higher quality research and more synthesisable data. This review is a starting point to produce more definitive conclusions about the effectiveness of interventions to prevent PPA in future systematic reviews.

## Figures and Tables

**Figure 1 jcm-11-06617-f001:**
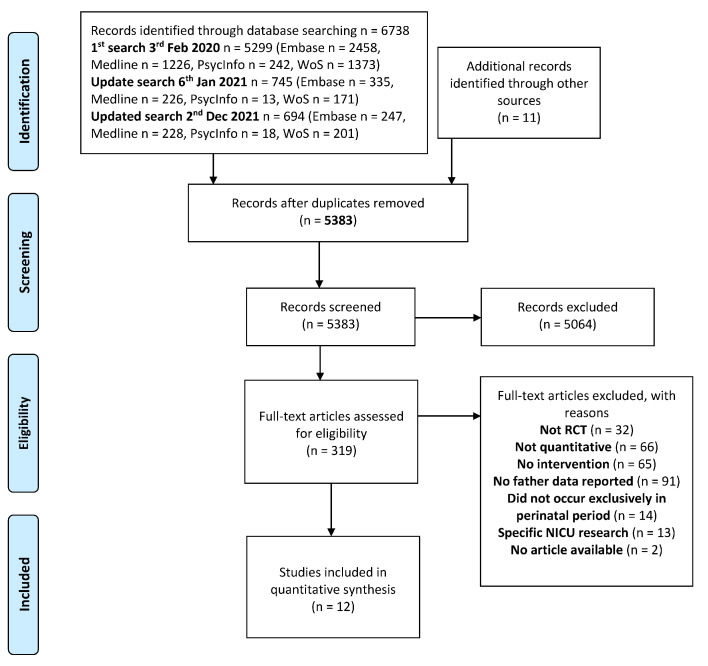
PRISMA (2009) Flow Diagram.

**Figure 2 jcm-11-06617-f002:**
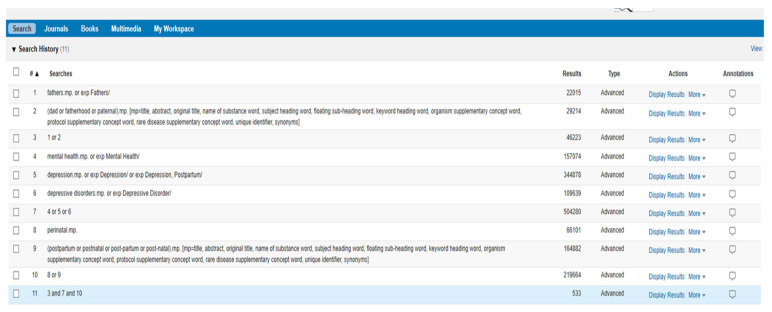
Draft electronic database search (Medline).

**Figure 3 jcm-11-06617-f003:**
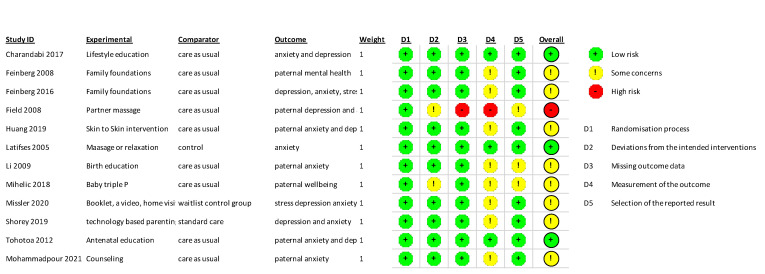
Risk of bias assessments [28,36,37,38,39,40,41,42,43,44,45,46].

**Figure 4 jcm-11-06617-f004:**
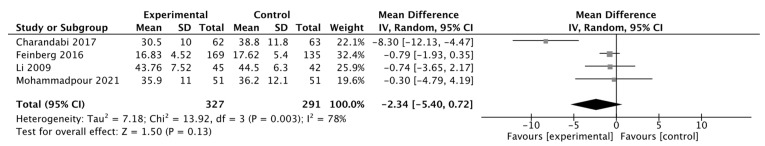
Trait anxiety forest plot [38,41,43,46].

**Figure 5 jcm-11-06617-f005:**
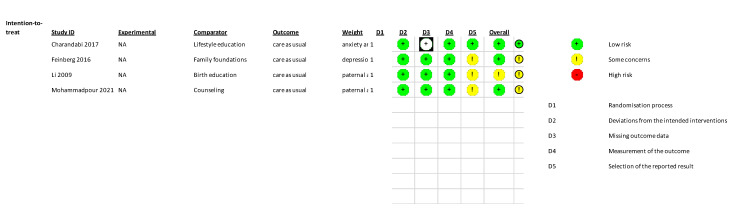
Risk of Bias assessments – trait anxiety outcomes [38,41,43,46].

**Figure 6 jcm-11-06617-f006:**
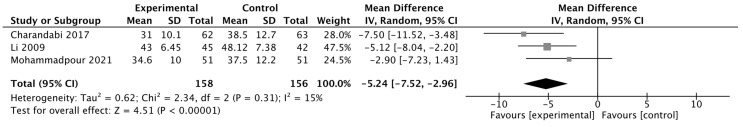
State anxiety forest plot [38,43,46].

**Figure 7 jcm-11-06617-f007:**
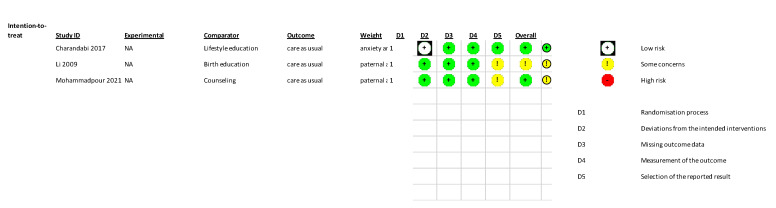
Risk of Bias assessments – state anxiety outcomes [38,43,46].

**Table 1 jcm-11-06617-t001:** PICOS Table.

Subject	Concepts
**Population**	Adult males:Studies with samples involving males over the age of 18 who are partners of pregnant women or with a baby up to one year of age.
**Intervention**	Any form of preventative or treatment intervention will be included, for example: Educational, pharmaceutical, parenting, therapeutic, psychosocial and practical skills training. Interventions must report the baseline and post-intervention outcomes of anxiety for fathers specifically.
**Comparator**	Any standard care provision or control arm activity:This includes standard care provision for prevention or treatment of adult mental health issues in the UK, as there is no standard care provision specific to paternal mental health.
**Outcomes**	Primary outcome:Reduction in reported anxiety scores using any validated assessment scales for measuring mental health. For example: Generalized Anxiety Disorder Severity Scale (GADSS), State-Trait anxiety inventory (STAI)Secondary outcome:If the necessary data are available, we will implement additional analysis of the following core outcomes; Self-assessed symptoms Satisfaction with intervention Parent-infant bonding Adverse events and suicidal thoughts Plus two additional outcomes which we have agreed provide insight into the effectiveness of interventions; 5. Timing of intervention; antenatal intervention vs. postnatal intervention, and/or antenatal to postnatal (transitional) interventions 6. Intervention delivery mode; face to face/groupwork/technology-based/couple-focussed

**Table 2 jcm-11-06617-t002:** Study characteristics.

First Author (Year)/(Ref Number)	Locality	Intervention Title (Type)	Intervention Summary	Comparator	Delivery Mode	N Used at Final Analysis	Delivery Time
Psychoeducational							
*Group-based couples*							
Feinberg (2016)/[41]	USA	Family foundations (parenting education program)	9 sessions: five classes before birth (3 h each) and four after birth (2 h each), focusing on co-parental conflict resolution and problem solving, communication, and mutual support strategies	Care as usual	Face to face	N = 608 (women = 304, men = 304) IG = 152 CG = 152	Antenatal to postnatal
Feinberg (2008)/[36]	USA	Family foundations (parenting education program)	8 sessions: consisting of four prenatal and four postnatal sessions, focusing on co-parental conflict resolution and problem solving, communication, and mutual support strategies	mailed a brochure about selecting quality childcare	Face to face	N = 304 (women = 152, men = 152) IG = 79 CG = 73	Antenatal to postnatal
Mihelic (2018)/[39]	Australia	Baby Triple P (parenting education program)	8 sessions—4 2 h groups sessions, 4 telephone consultations: 1. Positive Parenting, 2. Responding to your Baby, 3. Individual Survival Skills, 4. Partner Support, 5–8. Telephone Consultations	Care as usual	Face to face + Tele	N = 224 (women = 112, men = 112), IG = 55 CG = 57	Antenatal
Tohotoa (2012)/[28]	Australia	(antenatal education)	1 session: antenatal education session led by a male facilitator, followed by a six-week postnatal social support/education intervention consisting of education and support materials	Standard antenatal education	Face to face	N = 556 IG = 303 CG = 253	Antenatal to postnatal
*Individual couples*							
Missler (2020)/[45]	Netherlands	(psychoeducational parenting intervention)	(1) an information booklet; (2) an online video, (3) a prenatal home visit; and (4) a postnatal phone call	Waitlist control group	Face to face + Tele	N = 189 (women = 120, men = 69), IG = 31 CG = 38	Antenatal to postnatal
Shorey (2019)/[44]	Singapore	(technology-based supportive educational parenting program)	(1) a 30-min telephone-based antenatal educational session, (2) a 60-min telephone-based immediate postnatal educational session, and (3) a mobile health (mHealth) app follow-up educational session made available for 4 weeks postpartum.	Care as usual	Tele + App	N = 236 (women = 118, men = 118) IG = 59 CG = 59	Antenatal to postnatal
*Fathers only*							
Charandabi (2017)/[38]	Iran	(prenatal lifestyle-based education)	2 sessions: sleep health, nutrition, physical and sports activity, self- image and sexual problems, + weekly telephone counselling up to 6 weeks postnatal	Care as usual	Group Face to fac e+Tele	N = 125 IG = 62 CG = 63	Antenatal to postnatal
Li (2009)/[43]	Taiwan	Birth Education Program for Expectant Fathers Who Plan to Accompany Their Partners Through Labour (childbirth classes)	1 session: Labour and delivery, discussed the concerns of expectant fathers, and demonstrated how each expectant father could support and assist with his partner’s labour pain and relax himself	Care as usual	Individual Face to face	N = 87 IG n = 45 CG n = 42	Antenatal
Mohammadpour (2021)/[46]	Iran	Counselling sessions (4 weeks) to familiarize fathers with changes in pregnancy and their role in maternal and foetal health.	Effect of social support on mother and foetus during pregnancy, the role of fathers in supporting Mental health during pregnancy, anatomical, physio- logical, and hormonal changes Stages of foetus development during pregnancy, the effect of pregnant mothers’ nutrition and fathers’ attention to the feeding of their pregnant wives Childbirth preparation risk signs and symptoms during pregnancy and dealing with them, delivery process and stages, delivery methods	Care as usual	Group Face to face	N = 102 IG n = 51 CG n = 51	Antenatal
**Practical skills**							
Field (2008)/[37]	USA	(pregnancy massage therapy)	Single Session + DVD: instruction to massage twice per week, 16 weeks	Care as usual	Individual Couples Face to face + DVD	N = 114 (women = 57, men = 57) IG = 29 CG = 28	Antenatal
Huang (2019)/[40]	China	(paternal skin to skin contact)	1 session (30 min): provision of special caregiving training and explained the benefits of SSC	Care as usual	Individual Fathers only Face to face	N = 100 IG n = 50 CG n = 50	Postnatal
Latifses (2005)/[42]	USA	(1. partner massage) (2. relaxation)	Antenatal classes, 1 taught session: 1. Fathers were taught to massage their pregnant wives, provided with a handout, and instructed to massage their partner twice weekly for 5 weeks. 2. Both the father and mother were taught a 20-min relaxation program and instructed to listen to an audiotape of the program twice weekly for 5 weeks.	Care as usual	Individual Couples Face to face	N = 278 (women = 139, men = 139) IG1 = 47 IG2 = 46 CG n = 46	Antenatal

IG = Intervention Group; CG = Comparator Group.

## Data Availability

Not applicable.

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
