# Peer review of "The Effectiveness of Interventions for the Prevention or Treatment of Paternal Perinatal Anxiety: A Systematic Review"

_jcm, 2022, doi:10.3390/jcm11226617_

Round 1
Reviewer 1 Report
I have reviewed "The effectiveness of interventions for the prevention or treatment of paternal perinatal anxiety: a systematic review" by Fisher et al., the manuscript consists of 39 pages and is written in proper English (the authors are native speakers). The manuscript is a systematic review which follows PRISMA and it was properly conducted. I have no other comments but one: please put an information that the manuscript followed PRISMA in the abstract.
Author Response
Please see attachment - amendment has been highlighted on page 1.

Reviewer 2 Report
Thank you for the opportunity to review this work that evaluates interventions for the prevention or treatment of paternal perinatal anxiety. The issue is of high relevance considering that the present review addresses an important gap in the current literature which has focused more on these issues as they concern mothers to be rather, and rather neglecting the community of fathers. Authors do a good job in highlighting the importance of evaluating perinatal anxiety in fathers and implementing treatment programs to address this issue in a timely and efficient manner. Although the number of studies considered is still limited and methodological variability does not allow for robust conclusions, this systematic review represents an important first step towards raising awareness on the issue and orienting future research. I have some minor comments to the authors that I have included in an annotated copy of the manuscript (see attached) which I believe will improve the overall quality of the paper.

Author Response
Please see attachment - amendments have been highlighted.
